# A Molecular Signature of Circulating MicroRNA Can Predict Osteolytic Bone Disease in Multiple Myeloma

**DOI:** 10.3390/cancers13153877

**Published:** 2021-07-31

**Authors:** Aristea-Maria Papanota, Panagiotis Tsiakanikas, Christos K. Kontos, Panagiotis Malandrakis, Christine-Ivy Liacos, Ioannis Ntanasis-Stathopoulos, Nikolaos Kanellias, Maria Gavriatopoulou, Efstathios Kastritis, Margaritis Avgeris, Meletios-Athanasios Dimopoulos, Andreas Scorilas, Evangelos Terpos

**Affiliations:** 1Department of Clinical Therapeutics, School of Medicine, National and Kapodistrian University of Athens, 11528 Athens, Greece; ampapanota@med.uoa.gr (A.-M.P.); panosmalan@med.uoa.gr (P.M.); liakou@med.uoa.gr (C.-I.L.); johnntanasis@med.uoa.gr (I.N.-S.); nkanellias@med.uoa.gr (N.K.); mgavria@med.uoa.gr (M.G.); ekastritis@med.uoa.gr (E.K.); mdimop@med.uoa.gr (M.-A.D.); 2Department of Biochemistry and Molecular Biology, Faculty of Biology, National and Kapodistrian University of Athens, 15701 Athens, Greece; ptsiak@biol.uoa.gr (P.T.); chkontos@biol.uoa.gr (C.K.K.); mavgeris@med.uoa.gr (M.A.); 3Laboratory of Clinical Biochemistry-Molecular Diagnostics, Second Department of Pediatrics, School of Medicine, National and Kapodistrian University of Athens, “P. & A. Kyriakou” Children’s Hospital, 11527 Athens, Greece

**Keywords:** blood plasma, circulating miRNA, molecular biomarker, hematological malignancies, MMBD diagnosis, osteolysis, skeletal-related events, prognosis, qPCR, multiparametric model

## Abstract

**Simple Summary:**

Multiple myeloma bone disease (MMBD) is one of the most important complications of multiple myeloma with a great impact on quality of life. Recent advances in the field of imaging techniques provided clinicians with a variety of imaging modalities with high sensitivity for the diagnosis of MMBD. However, no circulating biomarkers are available to support the diagnosis of MMBD in cases where the results are inconclusive. The aim of our study was to investigate the clinical utility of 19 miRNAs implicated in osteoporosis in MMBD. Our results suggest that the levels of circulating let-7b-5p, miR-143-3p, miR-17-5p, miR-335-5p, and miR-214-3p (standalone or combined in multi-miRNA models) can effectively predict the presence of MMBD in newly diagnosed MM patients.

**Abstract:**

Background: Multiple myeloma bone disease (MMBD) constitutes a common and severe complication of multiple myeloma (MM), impacting the quality of life and survival. We evaluated the clinical value of a panel of 19 miRNAs associated with osteoporosis in MMBD. Methods: miRNAs were isolated from the plasma of 62 newly diagnosed MM patients with or without MMBD. First-strand cDNA was synthesized, and relative quantification was performed using qPCR. Lastly, we carried out extensive biostatistical analysis. Results: Circulating levels of let-7b-5p, miR-143-3p, miR-17-5p, miR-214-3p, and miR-335-5p were significantly higher in the blood plasma of MM patients with MMBD compared to those without. Receiver operating characteristic curve and logistic regression analyses showed that these miRNAs could accurately predict MMBD. Furthermore, a standalone multi-miRNA–based logistic regression model exhibited the best predictive potential regarding MMBD. Two of those miRNAs also have a prognostic role in MM since survival analysis indicated that lower circulating levels of both let-7b-5p and miR-335-5p were associated with significantly worse progression-free survival, independently of the established prognostic factors. Conclusions: Our study proposes a miRNA signature to facilitate MMBD diagnosis, especially in ambiguous cases. Moreover, we provide evidence of the prognostic role of let-7b-5p and miR-335-5p as non-invasive prognostic biomarkers in MM.

## 1. Introduction

Multiple myeloma (MM) is a hematological malignancy characterized by clonal plasma cell proliferation in the bone marrow. Monoclonal gammopathy of undetermined significance (MGUS) and smoldering multiple myeloma (sMM) are asymptomatic states preceding multiple myeloma and belong in a spectrum of disorders referred to as plasma cell dyscrasias. Symptomatic MM is characterized by end-organ damage as indicated by the acronym “CRAB” or by the presence of one or more of the recently introduced biomarkers of malignancy. Typical clinical manifestations of MM include hypercalcemia, renal impairment, anemia, and bone disease [1]. Myeloma bone disease (MMBD) is one of the most devastating complications of MM, with a great impact on patients’ survival and quality of life. About 80% of MM patients present bone disease in the form of osteolysis at diagnosis [2]. Whole-body, low-dose computed tomography (WBLDCT) is currently the standard imaging technique used to detect MMBD. Moreover, nowadays, additional imaging modalities such as whole-body magnetic resonance imaging (WBMRI) and positron emission tomography–computed tomography (PET/CT) are available, which can also assess the activity of the disease [3,4]. Despite the progress made in the field of MMBD diagnosis, there is an unmet need for accurate diagnostic biomarkers to facilitate the diagnosis of MMBD in ambiguous cases, thus guiding therapeutic decisions.

Normally, bone remodeling is a fine-tuned procedure that is crucial for the skeleton to sustain the mechanical load. The bone marrow microenvironment consists of various types of cells, including osteocytes, osteoblasts, osteoclasts, stromal cells, and immune cells. In non-pathological states, the finely tuned balance between osteoclast-dependent bone resorption and osteoblast-dependent bone formation orchestrates the bone remodeling process [5]. In MMBD, the interplay between bone marrow stromal cells (BMSCs) and malignant plasma cells, as well as the production of several endogenous metabolites by the latter, leads to increased osteoclast and decreased osteoblast activity, shifting the intrinsic balance towards bone destruction. The molecular background of MMBD is extremely complex as many signaling pathways and cell to cell crosstalk are implicated in this process [6,7,8].

MicroRNAs (miRNAs) are endogenous, single-stranded, small RNA molecules (approximately 21 nucleotides long) that regulate gene expression at the transcriptional and post-transcriptional levels. Mature miRNAs interact with 3′ untranslated regions (UTR) of mRNA target genes through regions of homology, found in their 5′ end, widely known as the seed region. The degree of complementarity between the seed region of the miRNA and the 3′ UTR of the target mRNA defines whether or not the corresponding target will be degraded or translationally repressed [9]. Consequently, those small RNA molecules have emerged as key regulators of gene expression, and therefore are implicated in a variety of normal biological processes, including bone remodeling [10,11,12]. Under that prism, a plethora of recent scientific evidence highlights the impact of miRNA deregulation on impaired bone remodeling that leads to bone disease of different etiology such as osteoporosis and MM [13,14,15,16]. Regarding multiple myeloma, scientific research proved that miRNAs serve as regulators of the major signaling pathways and also regulate the pro-inflammatory bone marrow microenvironment in MM, thus possessing a significant role in the pathophysiology of MMBD [17].

Although miRNAs are primarily localized in the cytoplasm, they can also be secreted in body fluids such as blood, urine, and saliva [18,19,20]. miRNAs are remarkably stable in human body fluids—similarly to circular RNAs [21,22]—and can be easily detected using conventional molecular techniques, thus representing promising candidate molecules to serve as non-invasive biomarkers in a variety of diseases [23]. Circulating miRNAs were first used as biomarkers for diffuse large B-cell lymphoma in 2008 [24], and since then, they are widely referred to in the literature as potential biomarkers for many hematological malignancies [12,25], similarly to other small non-coding RNAs such as tRNA fragments [26,27,28,29]. However, the role of circulating miRNAs as biomarkers for MMBD is not extensively studied. In this study, we evaluated, for the first time, the clinical utility of an established panel of 19 miRNAs associated with bone disease in osteoporosis as molecular markers, indicating MMBD in MM patients.

## 2. Materials and Methods

### 2.1. Study Design and Participants

Peripheral blood plasma samples were obtained from 10 healthy donors as well as 62 multiple myeloma (MM) patients with or without MMBD at the time of diagnosis. Additionally, bone marrow aspirate samples (BMA) were obtained from 30 out of the 62 MM patients. The study was conducted at the Department of Clinical Therapeutics (General Hospital of Athens “Alexandra”) of the National and Kapodistrian University of Athens, between 12/2017 and 7/2019. Patients that had already received any kind of treatment were excluded from our study. The presence of MMBD was assessed by WBLDCT, following the most recent International Myeloma Working Group (IMWG) recommendations [3]. A complete medical record that included patients’ survival data, important clinicopathological features, and data regarding the severity of bone disease, such as the number of osteolytic lesions and the presence of skeletal-related events (SREs), was created. The current study was approved by the scientific board of the General Hospital of Athens “Alexandra” and conducted according to the ethical principles of the 1964 Declaration of Helsinki and its later amendments. Written informed consent was obtained from each patient to provide a sample for research purposes.

### 2.2. Biological Material

Whole blood samples, as well as BMA samples, were collected in ethylenediaminetetraacetic acid (EDTA) containing tubes from participants enrolled in the current study.

Initially, blood samples were centrifuged at 1500× *g* for 10 min at 4 °C to remove blood cells from plasma samples. Subsequently, plasma samples were centrifuged at 1500× *g* at 4 °C for 10 min to deplete platelets. The resulting supernatant was transferred into new RNase-free tubes and then subjected to a second centrifugation at 17,000× *g* at 4 °C for 10 min to completely remove the remaining cellular debris, reducing the potential contamination from blood cells. The presence of hemolysis was assessed by quantifying the spectrophotometric absorbance of hemoglobin (414 nm) in the plasma samples. Samples exceeding the absorbance value of 0.2 were considered hemolyzed and excluded from downstream analysis.

Processing the BMA samples, we used Ficoll-Paque to isolate mononuclear cells. Next, CD138+ plasma cells were positively selected using magnetic beads coated with an anti-CD138 antibody (Miltenyi Biotec, Bergisch Gladbach, Germany). Total RNA extraction from CD138+ cell pellets was performed using TRI Reagent^®^ (Molecular Research Center, Inc., Cincinnati, OH, USA). Finally, RNA pellets from bone marrow plasma cells were dissolved in THE RNA Storage Solution (Ambion™, Thermo Fisher Scientific Inc., Waltham, MA, USA).

All samples were immediately stored at −80 °C to preserve RNA integrity until further processing.

### 2.3. miRNA Isolation and First-Strand cDNA Synthesis

The small RNA fraction was isolated from an initial volume of 300 μL fresh-frozen plasma samples, using NucleoSpin™ miRNA Plasma kit (Macherey-Nagel GmbH & Co. KG, Duren, Germany) according to the manufacturer’s guidelines. Subsequently, the isolated RNA was dissolved in 20 μL of RNA Storage Solution (Life Technologies Ltd., Carlsbad, CA, USA) and stored at −80 °C until further processing. Prior to isolation, 25 fmol of synthetic cel-mir-39-3p was added to each sample, serving as an exogenous reference control which was used to monitor the sample-to-sample variation during the isolation procedure and to perform the downstream normalization of miRNA levels. The determination of RNA yield was carried out using Qubit fluorometer (Invitrogen™, Thermo Fisher Scientific Inc.). Total RNA was also extracted from the CD138+ plasma cells positively selected from the BMA samples.

Next, a universal reverse transcription (RT) reaction was performed to convert all miRNA into complementary DNA (cDNA) according to the manufacturer’s instructions. In brief, prior to RT, a poly(A) tail was added at the 3′ end of the miRNA template. Finally, cDNA was synthesized using an oligo-dT–primer with a 3′ degenerate anchor and a 5′ universal tag. 

### 2.4. Circulating miRNA Detection Using Quantitative PCR (qPCR)

Following RNA polyadenylation and first-strand cDNA synthesis, the levels of an established panel of 19 miRNAs (let-7b-5p, miR-17-5p, miR-19b-3p, miR-29b-3p, miR-31-5p, miR-127-3p, miR-133b, miR-141-3p, miR-143-3p, miR-144-5p, miR-152-3p, miR-188-5p, miR203a, miR-214-3p, miR-320a-3p, miR-335-5p, miR-375-3p, miR-550a-3p, miR-582-5p) associated with bone disease in osteoporosis of different etiologies were quantified in plasma using osteomiR qPCR assay (TAmiRNA GmbH). Briefly, qPCR was carried out in ready-to-use plates pre-coated with specific LNA-enhanced forward and reverse primers for the corresponding miRNA targets. The amplification and detection of target miRNAs were conducted in an ABI 7500 Fast Real-Time PCR System (Applied Biosystems, Foster City, CA, USA) based on the miGreen chemistry, according to the manufacturer’s guidelines.

Subsequently, we applied the comparative C_q_ method for the relative quantification of target miRNAs in the plasma samples from MM patients and normal controls, as well as in the CD138+ plasma cells from bone marrow of MM patients. For this purpose, we used reference transcripts to normalize qPCR for the amount of RNA used in RT reactions. Spiked-in cel-miR-39-3p was used for the normalization of circulating miRNA levels, whereas small nucleolar RNA C/D box 48 *SNORD48* (*RNU48*) was exploited as an endogenous reference for the normalization of intracellular miRNA expression levels of CD138+ plasma cells. Assessment of hemolysis in plasma samples was accomplished by calculating the miR-451a to miR-23a-3p ratio as previously described [30]. The normalized results for each target miRNA were presented as relative quantification units (RQU).

### 2.5. Biostatistical Analysis

Statistical analysis of our data was performed using the IBM SPSS Statistics 26 software (IBM Corp.). Due to the non-Gaussian distribution of miRNA levels, the non-parametric Mann–Whitney *U* test was used to assess the significance of the differences observed in the miRNA levels between MM patients with and without MMBD.

The clinical ability of the significantly differentiated miRNAs to discriminate MMBD from non-MMBD patients was evaluated by a receiver operating characteristic (ROC) and logistic regression analysis. Bootstrap logistic regression analysis using 1000 bootstrap samples was carried out for internal validation. The area under the curve (AUC) for each miRNA was calculated and compared to the rest regarding its ability to accurately distinguish MMBD from non-MMBD patients. Next, we developed multiple multivariate logistic regression models by combining the miRNAs that presented the higher discriminatory potential to successfully predict MMBD. The evaluation and comparison of these models were accomplished by ROC analysis, in which predicted probabilities of developing MMBD were used as input values.

Finally, potential associations between target miRNAs expression status and survival of MM patients were evaluated by Kaplan–Meier, using the log-rank test, as well as Cox regression analysis and bootstrap Cox regression using 1000 bootstrap samples for internal validation, as previously described [31,32]. The binary classification of MM patients as high or low expression of the corresponding miRNAs was accomplished by X-tile algorithm, which provides an optimal cut-off point using the minimum *p* value approach [33]. The significance threshold of all statistical tests was set at a probability value of less than 0.050 (*p* < 0.050).

## 3. Results

### 3.1. Baseline Clinical Characteristics of MM Patients

The present study included a total of 62 newly diagnosed MM patients. The study cohort consisted of 35 (56.5%) male and 27 (43.5%) female patients. The median age at the time of diagnosis was 62 years, ranging from 35 to 90 years. At diagnosis, osteolytic lesions were observed in 35 out of 62 patients (56.5%), whereas the rest 27 (43.5%) had no signs of MMBD. Moreover, within the subgroup of MMBD patients, 20 out of 35 (57.1%) presented SREs at diagnosis. The prognostic stratification of MM patients was carried out using the Revised International Staging System (R-ISS). In detail, 18 patients (29%) were R-ISS stage I, 25 (40.3%) in R-ISS stage II, 12 (19.4%) in R-ISS stage III, while for the remaining 7 (11.3%), data were unavailable. The baseline clinical characteristics of MM patients are summarized in Table 1.

### 3.2. Circulating miRNAs Can Distinguish MM Patients with Osteolytic Bone Disease

The analysis of the obtained data unveiled five circulating miRNAs, namely, let-7b-5p (*p* = 0.034), miR-143-3p (*p* = 0.021), miR-17-5p (*p* = 0.025), miR-214-3p (*p* = 0.004), and miR-335-5p (*p* = 0.022), that were significantly higher in the plasma samples of MMBD patients than in the plasma samples of non-MMBD patients (Figure 1) and those of normal controls (Appendix A).

To further investigate whether the levels of circulating let-7b-5p, miR-143-3p, miR-17-5p, miR-214-3p, and miR-335-5p were able to distinguish patients with MMBD from those without, we performed ROC and logistic regression analyses. Univariate logistic regression analysis unveiled that except for miR-17-5p, the remaining four miRNAs can accurately predict the presence of MMBD (Table 2). More specifically, overexpression of miR-143-3p in the plasma from MM patients was associated with an almost two-fold increased probability of MMBD at diagnosis. Similarly, elevated plasma levels of let-7b-5p, miR-214-3p, and miR-335-5p were associated with a three-fold increased probability of presenting MMBD (Table 2). Interestingly, a similar trend was observed for each of these miRNAs in CD138+ plasma cells that were positively selected from BMA samples of MMBD patients, compared to their intracellular expression levels in CD138+ plasma cells from BMA samples of MM patients without bone disease (Appendix A).

These results were further confirmed by ROC analysis (Figure 2), which also indicated that all five miRNAs could effectively detect MMBD within the cohort of MM patients. The best discriminatory ability was observed for miR-214-3p, followed by miR-143-3p, miR-335-5p, miR-17-5p, and let-7b-5p. Although the aforementioned miRNAs can predict osteolytic bone disease in MM patients, our results failed to establish any significant association between their levels in plasma and the severity of MMBD, as indicated by the number of osteolytic lesions and the presence of SREs. Moreover, no significant correlation was observed between the circulating levels of these miRNAs in plasma and serum alkaline phosphatase (ALP) levels.

### 3.3. Construction and Evaluation of an MMBD-Predictive miRNA Model

Multiple logistic regression models were developed and used to estimate the probability of MM patients to present MMBD at diagnosis. These models included plasma levels of let-7b-5p, miR-143-3p, miR-214-3p, and miR-335-5p as variables, which have been validated in both logistic regression and ROC analyses as the most prominent molecular predictors of MMBD. To identify the most robust model in terms of predicting MMBD, we tested all the available 2-, 3-, and 4-miRNA combinations. The evaluation of the regression models was performed by ROC analysis, using the predicted probabilities of MM patients to be diagnosed with MMBD as input and comparing the corresponding AUC as described in “Materials and Methods”.

By implementing all the available 2-miRNA combinations, we observed that the most effective 2-miRNA model in predicting MMBD consisted of miR-214-3p and miR-335-5p. The probabilities of MMBD occurrence in MM patients were calculated by the 2-miRNA logit model, logit(P) = 1.85 × log(miR-214-3p) + 0.53 × log(miR-335-5p) + 0.61, and subsequently used to construct a ROC curve (Figure 3a). Next, we combined the three most effective miRNA predictors, namely, miR-214-3p, miR-335-5p, and let-7b-5p, to build a 3-miRNA logit model [logit(P) = 0.86 × log(let-7b-5p) + 1.74 × log(miR-214-3p) + 0.18 × log(miR-335-5p) − 3.71]. This combination provides a 3-miRNA model characterized by a slightly better predictive potential compared to the 2-miRNA model (Figure 3b). Finally, the combination of all 4 miRNAs into a standalone model [logit(P) = 0.87 × log(let-7b-5p) + 1.42 × log(miR-214-3p) − 0.29 × log(miR-335-5p) + 0.51 × log(miR-143-3p) − 3.68] failed to provide an increased predictive potential compared to 2- and 3-miRNA models (Figure 3c). The evaluation of the proposed models using ROC analysis revealed minor differences regarding their ability to predict MMBD in MM patients. However, the best predictive model consisted of let-7b-5p, miR-214-3p, and miR-335-5p. Based on the predicted probabilities derived as output from the 3-miRNA model, a probability cutoff value of 0.37 corresponds to increased diagnostic sensitivity (sensitivity: 96%, specificity: 55%), while a probability cutoff value of 0.70 corresponds to increased diagnostic specificity (sensitivity: 54%, specificity: 90%).

### 3.4. Evaluating the Prognostic Role of the MMBD-Specific miRNAs in MM

To study potential associations between both overall survival (OS) and progression-free survival (PFS) of MM patients with the levels of circulating let-7b-5p, miR-143-3p, miR-17-5p, miR-214-3p, and miR-335-5p, we categorized them as high or low expression as described in the “Materials and Methods” section. The obtained cutoff values were 86.2 RQU for let-7b-5p (equal to the fifty-third percentile), 0.4 RQU for miR-143-3p (equal to the thirty-fourth percentile), 6.1 RQU for miR-17-5p (equal to the fifty-third percentile), 0.42 RQU for miR-214-3p (equal to the thirty-seventh percentile) and 0.97 RQU for miR-335-5p (equal to the thirty-eighth percentile). Of the total 62 MM patients, 1 was excluded from survival analysis due to missing follow-up data. From the rest 61 MM patients, 11 (18%) patients died, and 18 (29.5%) presented disease progression during the follow-up period. The estimated median OS was 24 months (range: 6.0–32.0), while the median PFS was 20 months (range: 3.0–31.0). In univariate Cox regression analysis, lower plasma levels of let7b-5p and miR-335-5p revealed a significantly increased risk for disease progression (Table 3). Moreover, lower plasma levels of let-7b-5p and miR-335-5p retained their adverse prognostic significance independently of the established clinicopathological parameters such as patients’ age, R-ISS stage, B2M, and LDH (Table 3). In accordance with these results, Kaplan–Meier curves illustrated that MM patients with low let-7b-5p and/or miR-335-5p plasma levels had significantly shorter PFS (Figure 4a,b, respectively).

Finally, univariate Cox regression analysis (Table 4) unveiled a significant decrease in OS for MM patients with lower levels of circulating let-7b-5p that was further confirmed by Kaplan–Meier survival analysis (Figure 5).

## 4. Discussion

In the era of individualized and precision medicine, robust molecular markers are required to ameliorate therapeutic decisions [34,35]. The integration of diagnostic, prognostic, and predictive molecular markers revolutionizes the field of modern oncology by providing tailored therapeutic management [36,37,38]. As already mentioned, miRNAs regulate gene expression at the transcriptional and post-transcriptional levels. Thus, they are implicated in several biological procedures, and their deregulation can lead to a variety of diseases, especially malignancies [39]. Circulating miRNAs are carried in body fluids through microvesicles, exosomes, or are associated with proteins, presenting remarkable stability within body fluids. They are easily accessible by minimally invasive techniques and can be quantified as candidate biomarkers for a wide spectrum of diseases [40,41,42]. As a result, circulating miRNAs can be used either standalone or combined in miRNA-panels as molecular markers able to support diagnosis and guide therapeutic decisions [23,43]. Although miRNAs are extensively investigated as molecular markers related to prognosis in MM, their potential clinical value as biomarkers for MMBD has been neglected [44,45].

Recently, scientific research proved that miRNAs play an important role both in osteoblastogenesis and osteoclastogenesis, and their deregulation is implicated in bone disease of any cause and, more specifically, in MMBD [17]. Our study identified five miRNAs that are differentially expressed in the plasma of patients with MMBD compared to those without. More specifically, expression of let-7b-5p, miR-143-3p, miR-17-5p, miR-335-5p, and miR-214-3p was significantly higher in the plasma of patients with MMBD. Although those molecules exhibited significantly different expressions between patients with and without MMBD, unfortunately, no correlation was observed between their expression and the severity of MMBD or the presence of skeletal-related events (SREs). A similar trend was also observed with regard to the intracellular levels of these five miRNAs, once their expression in CD138+ plasma cells of BMA samples was compared between MMBD patients and MM patients without bone disease. These results indicate that the levels of these circulating miRNAs in plasma could reflect their levels in malignant cells of bone marrow. As a result, miRNA-based diagnostics may provide valuable clinical information regarding osteolytic bone disease in MM.

Following this observation, we investigated the ability of the aforementioned molecules to discriminate patients with and without MMBD by logistic regression and ROC analysis. ROC analysis revealed that all these molecules can distinguish between MMBD and non-MMBD patients. Furthermore, miR-214-3p displayed the best discriminatory ability with an AUC value of 0.73. Moreover, miR-143-3p, miR-335-5p, miR-17-5p, and let-7b-5p displayed an AUC value of 0.68, 0.68, 0.67, and 0.66, respectively. Based on these observations, we developed several MMBD-predictive models, incorporating the expression of the aforementioned miRNAs as important variables. The best predictive model comprised let-7b-5p, miR-214-3p, and miR-335-5p.

Unsurprisingly, miR-214-3p expression is highlighted as the most efficient predictor of MMBD (Figure 2). According to the literature, miR-214-3p is implicated in bone disease by suppressing osteoblast differentiation and promoting osteoclastogenesis. In detail, miR-214-3p directly targets activating transcription factor 4 (*ATF4*), osterix, and fibroblast growth factor 1 (*FGFR1*), all of them characterized by an established role in osteoblast formation, thus leading to impaired osteoblastogenesis [11,46,47]. On the other hand, miR-214-3p also participates in the regulation of osteoclastogenesis. Zhao et al. reported that miR-214-3p is upregulated during osteoclastogenesis. More specifically, miR-214-3p targets PTEN and regulates osteoclastogenesis through the PI3K/Akt pathway [48]. Based on those observations, in 2016, Hao et al. investigated its role as a biomarker in MMBD. This study indicated that miR-214-3p is higher in the plasma of patients with MMBD compared to those without and thus could serve as a diagnostic biomarker able to support the diagnosis of MMBD. Our results are consistent with those findings since we showed that miR-214-3p is higher in MMBD patients and can discriminate patients with MMBD from those without. Furthermore, Hao et al. reported that miR-214-3p expression correlates with the severity of bone disease; however, such a relationship did not emerge in our study.

Interestingly, our study indicates, for the first time, a potential association of miR-143-3p, let-7b-5p, miR-335-5p, and miR-17-5p with MMBD. MiR-143-3p is an miRNA with proven clinical significance in osteosarcoma [49]. There are conflicting data regarding the role of miR-143-3p overexpression or downregulation in bone formation and destruction. Overexpression of miR-143-3p was shown to suppress osteogenesis by directly targeting and downregulating osterix [50], whereas other studies indicate that miR-143-3p overexpression leads to increased bone formation and decreased bone destruction [51,52]. The results of our study indicated that high levels of miR-143-3p are observed in patients presenting MMBD, distinguishing the latter from those without, thus possessing a role as a putative MMBD-predictive marker in MM patients. Regarding let-7 miRNA family members, a plethora of evidence supports a potential key role for several members of this family in bone formation. Overexpression of let-7 miRNA family members, such as let-7a-5p, inhibits osteogenesis by directly targeting TGFBR1, a serine/threonine kinase receptor implicated in TGF-β signaling [53,54]. These results are in line with the presented results, as we observed that elevated levels of let-7b-5p are indicative of MMBD. Regarding miR-335-5p, although basic research provides evidence that it promotes osteogenesis by targeting and downregulating DKK1 [55], studies in osteoporosis-related bone disease correlate high levels of circulating miR-335-5p with osteoporotic bone disease with fractures [56,57]. These results are consistent with the results of our study that support the increase of miR-335-5p in the blood plasma of MMBD patients. Finally, there are several reports in the literature supporting the inhibitory role of miR-17-5p in osteogenesis. Several targets of miR-17, including SMAD5 and BMP2, both involved in the TGFβ signaling pathway, are negatively regulated by miR-17-5p to suppress osteogenesis. [58,59,60]. In a study evaluating the levels of specific circulating miRNAs in blood samples from osteoporosis patients, circulating miR-17-5p levels were higher, similar to the results presented in our study regarding MMBD [61].

Regarding the prognostic role of the investigated miRNAs, our study reported that downregulation of both let-7b-5p and miR-335-5p is associated with significantly shorter PFS (Figure 4) and increased risk of disease progression independently of other established prognostic factors in MM (Table 3). Moreover, downregulation of let-7b-5p was also found to be an adverse prognostic factor regarding OS in MM patients (Figure 5). This is not the first time that those molecules are associated with MM prognosis. Let-7b-5p is an established tumor suppressor in MM, which acts by targeting and downregulating, the receptor of insulin-like growth factor (IGF1R) and the oncogene MYC [62,63]. In accordance with the results of the current study, lower plasma levels of circulating let-7b-5p were found to be associated with shorter OS [64]. To the best of our knowledge, the biological role of miR-335-5p in MM has not been extensively studied. A recent report indicates a potential tumor suppressor role for miR-335-5p in MM through the downregulation of IGF1R [65]. A potential tumor suppressor role for miR-335-5p in MM is also supported by the results of the presented study, in which MM patients with lower plasma levels of miR-335-5p had a significantly worse prognosis compared to those with higher miR-335-5p expression.

Although our study identifies several circulating miRNAs associated with osteolytic bone disease in MM for the first time, it is characterized by some limitations that need to be addressed. The main drawback of the study is the lack of a sufficient number of MMBD patients and controls. This fact may lower the significance level of the presented findings regarding the MMBD-predictive role of the presented miRNAs as well as their prognostic value as non-invasive tumor markers in MM. Future studies should be conducted to confirm the actual role of these miRNAs, specifically in MM and MMBD.

## 5. Conclusions

In summary, our study indicates that increased levels of circulating let-7b-5p, miR-143-3p, miR-17-5p, miR-335-5p, and miR-214-3p can effectively predict (standalone and/or combined) the occurrence of MMBD in MM patients. Moreover, we provided sufficient evidence regarding the prognostic role of some of those miRNAs as potential non-invasive tumor markers in MM. To the best of our knowledge, this study provides, for the first time, evidence supporting the potential role of specific miRNAs in MMBD, setting the basis for more research in this topic able to elucidate the mechanisms and the clinical utility of those biomarkers in MMBD and MM prognosis.

## Figures and Tables

**Figure 1 cancers-13-03877-f001:**
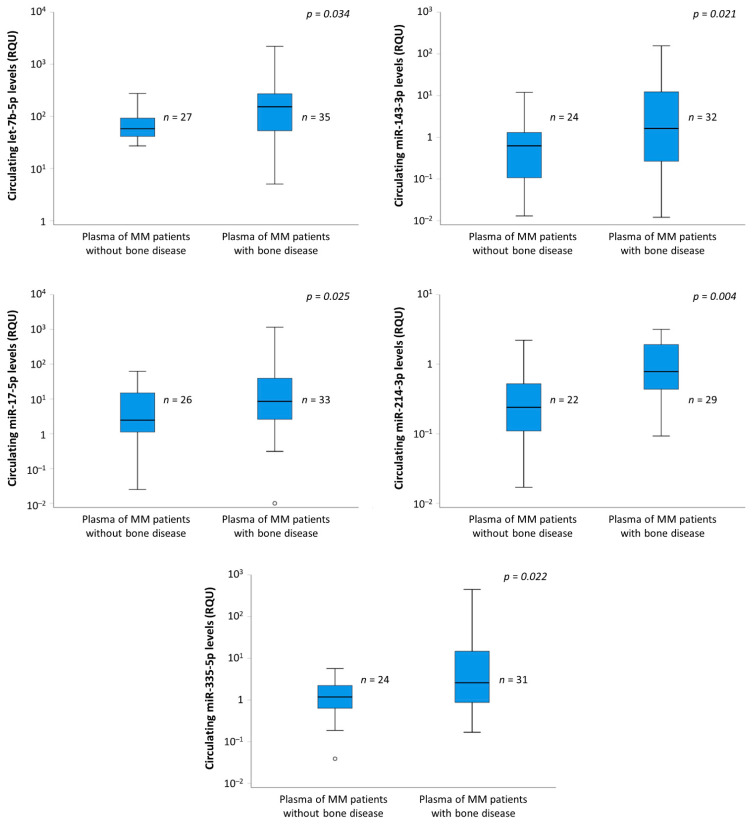
Comparison of let-7b-5p, miR-143-3p, miR-17-5p, miR-214-3p, and miR-335-5p levels in plasma of MM patients with and without MMBD. The expression of all miRNAs was higher in MM patients with MMBD compared to those without MMBD.

**Figure 2 cancers-13-03877-f002:**
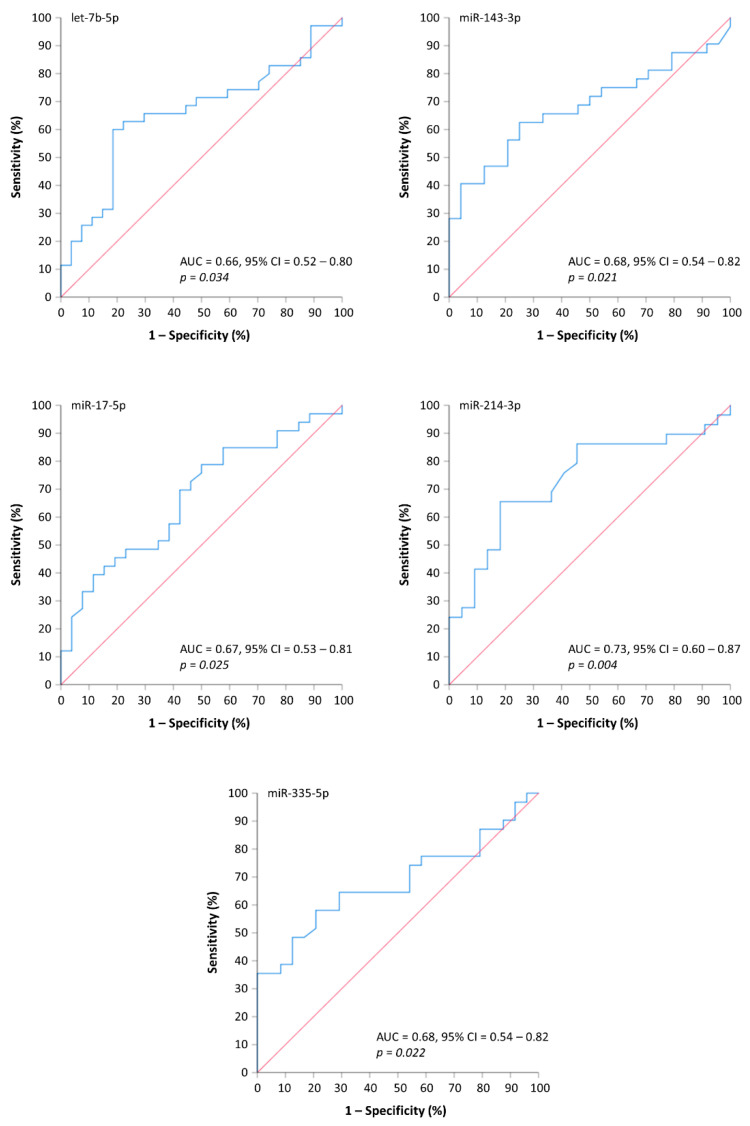
Receiver operating characteristic (ROC) curves based on let-7b-5p, miR-143-3p, miR-17-5p, miR-214-3p, and miR-335-5p levels in plasma. All miRNAs can distinguish MM patients with MMBD from those without MMBD.

**Figure 3 cancers-13-03877-f003:**
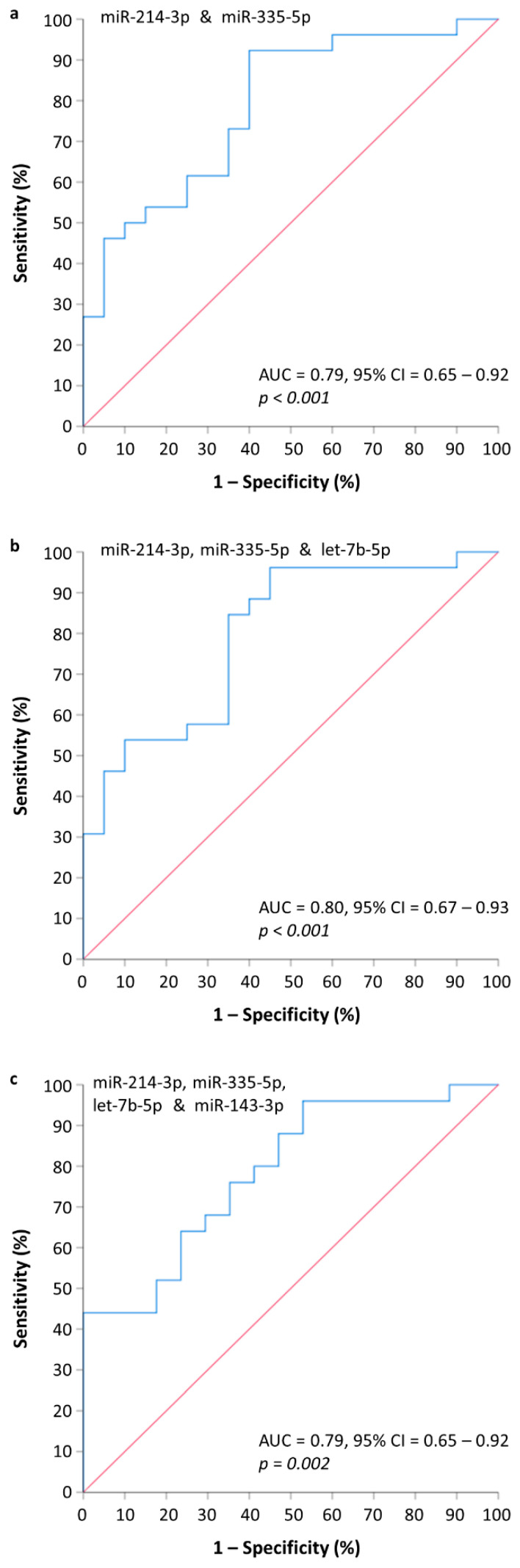
Receiver operating characteristic (ROC) curves for combinatorial models of (**a**) miR-214-3p and miR-335-5p; (**b**) miR-214-3p, miR-335-5p and let-7b-5p; (**c**) miR-214-3p, miR-335-5p, let-7b-5p, and miR-143-3p. The combination of miR-214-3p, miR-335-5p, and let-7b-5p into a standalone model constitutes the most prominent predictor of MMBD in MM patients.

**Figure 4 cancers-13-03877-f004:**
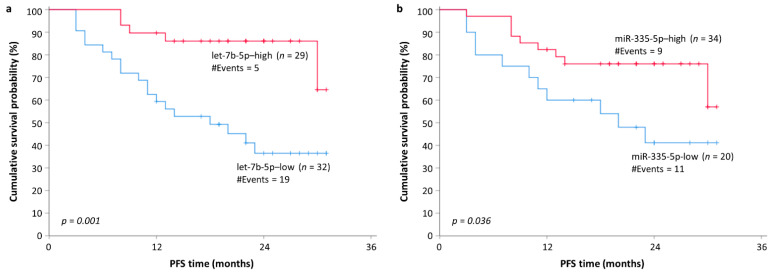
Kaplan–Meier survival curves for progression-free survival (PFS) of MM patients. Reduced plasma levels of (**a**) let-7b-5p and (**b**) miR-335-5p were shown as potentially unfavorable molecular markers of prognosis in MM.

**Figure 5 cancers-13-03877-f005:**
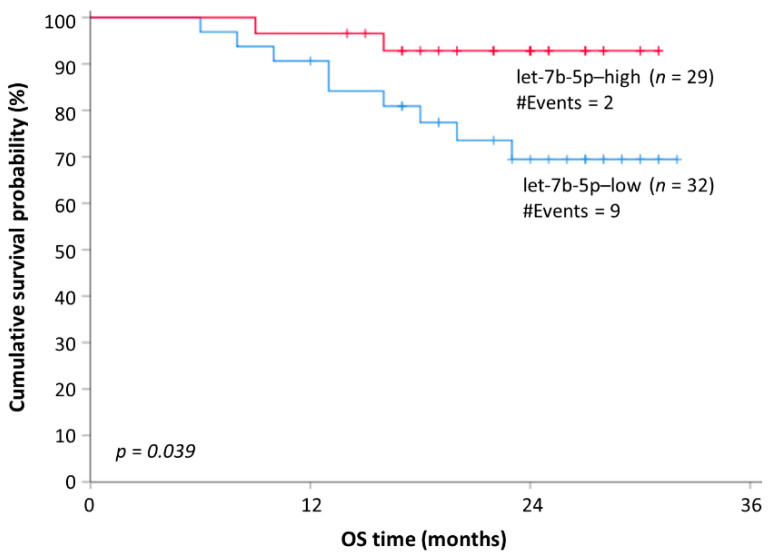
Kaplan–Meier survival curves for overall survival (OS) of MM patients. Reduced plasma levels of let-7b-5p were shown as potentially unfavorable molecular markers of prognosis in MM.

**Table 1 cancers-13-03877-t001:** Baseline clinical characteristics of MM patients.

Variable	Median (Range)
Age (years)	62 (35–90)
OS ^1^ (months)	24 (6–32)
PFS ^2^ (months)	20 (3–31)
	**Number of Patients (%)**
Gender	
Male	35 (56.5%)
Female	27 (43.5%)
MM type	
IgA	17 (27.5%)
IgG	35 (56.5%)
IgD	2 (3.2%)
κLC	2 (3.2%)
λLC	2 (3.2%)
NSMM ^3^	4 (6.4%)
ISS ^4^	
I	23 (37.1%)
II	15 (24.2%)
III	23 (37.1%)
Unavailable data	1 (1.6%)
Revised ISS ^4^ (R-ISS)	
I	18 (29.0%)
II	25 (40.3%)
III	12 (19.4%)
Unavailable data	7 (11.3%)
B2M ^5^	
≤5.5 mg/L	39 (62.9%)
>5.5 mg/L	23 (37.1%)
LDH ^6^	
Normal (≤225 U/L)	49 (79.0%)
Elevated (>225 U/L)	13 (21.0%)
ALP ^7^	
Normal (≤129 U/L)	59 (95.2%)
Elevated (>129 U/L)	3 (4.8%)
Primary treatment of MM	
Bortezomib-based	60 (96.8%)
IMiD-based ^8^	2 (3.2%)
HDM-ASCT ^9^	
Yes	38 (61.3%)
No	24 (38.7%)
MMBD ^10^	
Yes	35 (56.5%)
No	27 (43.5%)
SREs ^11^ (out of the 35 MMBD cases)	
Yes	20 (57.1%)
No	15 (42.9%)
BP ^12^ treatment	
Yes	35 (56.5%)
No	23 (37.0%)
Unavailable data	4 (6.5%)

^1^ Overall survival. ^2^ Progression-free survival. ^3^ Non-secretory multiple myeloma. ^4^ International Staging System. ^5^ β_2_ microglobulin (B2M). ^6^ Lactate dehydrogenase (LDH). ^7^ Alkaline phosphatase. ^8^ Immunomodulatory drug-based. ^9^ Autologous stem cell transplantation following high-dose melphalan. ^10^ Multiple myeloma bone disease. ^11^ Skeletal-related events (percentages regarding the MMBD cohort). ^12^ Bisphosphonate.

**Table 2 cancers-13-03877-t002:** Univariate logistic regression for predicting the presence of MMBD, based on the levels of let-7b-5p, miR-143-3p, miR-17-5p, miR-214-3p, and miR-335-5p in plasma.

Covariate	OR ^1^	95% CI ^2^	*p* Value ^3^	BCa Bootstrap 95% CI ^2^	Bootstrap *p* Value ^3^
let-7b-5p levels	3.13	1.04–9.44	*0.043*	1.86–14.30	*0.044*
miR-143-3p levels	1.86	1.07–3.23	*0.029*	1.14–4.13	*0.020*
miR-17-5p levels	1.89	1.01–3.54	*0.048*	1.07–4.62	0.052
miR-214-3p levels	3.07	1.15–8.21	*0.025*	1.28–20.49	*0.032*
miR-335-5p levels	3.24	1.23–8.51	*0.017*	1.57–10.70	*0.004*

^1^ Odds ratio, estimated from the logistic regression model. ^2^ Confidence interval of the estimated OR. ^3^ Statistically significant *p* values are shown in italics.

**Table 3 cancers-13-03877-t003:** Cox regression analysis regarding the prognostic potential of circulating let-7b-5p and miR-335-5p status for the PFS of MM patients.

	Variable (Tested vs. Control)	HR ^1^	95% CI ^2^	*p* Value ^3^	BCa Bootstrap 95% CI ^2^	Bootstrap *p* Value ^3^
**Univariate analysis**	Age	1.02	0.98–1.06	0.26	0.82–2.01	0.30
	R-ISS ^4^	1.58	0.89–2.83	0.12	0.93–2.88	0.097
	B2M ^5^ (>5.5 mg/L vs. ≤5.5 mg/L)	2.18	0.98–4.88	0.056	1.00–5.21	*0.045*
	LDH ^6^ (elevated vs. normal)	1.21	0.48–3.10	0.69	0.37–3.00	0.70
	let-7b-5p (high vs. low)	0.22	0.081–0.59	*0.003*	0.074–0.52	*0.004*
	miR-335-5p (high vs. low)	0.41	0.17–0.98	*0.044*	0.15–0.97	*0.027*
**Multivariate analysis ^7^**	Age	1.02	0.98–1.07	0.38	0.97–1.10	0.49
	R-ISS ^4^	1.23	0.42–3.54	0.71	0.32–5.64	0.70
	B2M ^5^ (>5.5 mg/L vs. ≤5.5 mg/L)	1.11	0.23–5.26	0.90	0.12–8.33	0.91
	LDH ^6^ (elevated vs. normal)	0.59	0.17–2.00	0.40	0.15–1.75	0.40
	let-7b-5p (high vs. low)	0.25	0.078–0.82	*0.022*	0.082–0.51	*0.011*
	Age	1.04	0.98–1.09	0.21	0.96–1.17	0.27
	R-ISS ^4^	1.06	0.33–3.33	0.93	0.20–6.55	0.93
	B2M ^5^ (>5.5 mg/L vs. ≤5.5 mg/L)	2.61	0.51–13.29	0.25	0.26–34.81	0.30
	LDH ^6^ (elevated vs. normal)	0.67	0.19–2.43	0.56	0.13–2.31	0.54
	miR-335-5p (high vs. low)	0.31	0.11–0.85	*0.024*	0.098–0.55	*0.021*

^1^ Hazard ratio, estimated from Cox proportional hazard regression model. ^2^ Confidence interval of the estimated HR. ^3^ *p* value (significant *p* values shown in italics). ^4^ Revised International Staging System. ^5^ β_2_ microglobulin (B2M). ^6^ Lactate dehydrogenase (LDH). ^7^ Multivariate models regarding PFS were adjusted for let-7b-5p or miR-335-5p and patients’ age, R-ISS stage, B2M, and LDH.

**Table 4 cancers-13-03877-t004:** Cox regression analysis regarding the prognostic potential of circulating let-7b-5p status for the OS of MM patients.

Variable (Tested vs. Control)	HR ^1^	95% CI ^2^	*p* Value ^3^	BCa Bootstrap 95% CI ^2^	Bootstrap *p* Value ^3^
Age	1.02	0.97–1.08	0.46	0.97–1.08	0.38
R-ISS ^4^	2.98	1.13–7.86	*0.028*	1.54–8.76	*0.002*
B2M ^5^ (>5.5 mg/L vs. ≤5.5 mg/L)	3.43	1.00–4.88	*0.049*	0.96–16.44	*0.017*
LDH ^6^ (elevated vs. normal)	2.04	0.59–6.98	0.26	0.40–7.31	0.21
let-7b-5p (high vs. low)	0.23	0.049–1.060	0.059	0.014–0.75	*0.025*

^1^ Hazard ratio, estimated from Cox proportional hazard regression model. ^2^ Confidence interval of the estimated HR. ^3^ Significant *p* values are shown in italics. ^4^ Revised International Staging System. ^5^ β_2_ microglobulin. ^6^ Lactate dehydrogenase.

## Data Availability

The data presented in this study are available on request from the corresponding author. The data are not publicly available due to ethical issues.

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
