# Peer review of "A Molecular Signature of Circulating MicroRNA Can Predict Osteolytic Bone Disease in Multiple Myeloma"

_cancers, 2021, doi:10.3390/cancers13153877_

Round 1
Reviewer 1 Report
The study by Papanota and colleagues addresses important and still unmet clinical issues. In particular, they investigated circulating miRNAs that could predict osteolytic bone disease in multiple myeloma.
As stated by the Authors, the main drawback of the study is the limited number of patients investigated. In this regard, I suggest presenting in the manuscript only the 62 sample actually investigated, as the information on the 2 samples with haemolysis and not considered for the analyzes could be misleading.
In addition, information concerning the expression level of investigated miRNAs in serum from healthy donors could be very interesting.
Minor comments:
The information in the text and in the corresponding tables is quite redundant; in my opinion the paragraphs could be shortened.
In figure 4b, the Kaplan Meier reported only 54 out of the 61 patients with clinical data, why?
Author Response
- As stated by the Authors, the main drawback of the study is the limited number of patients investigated. In this regard, I suggest presenting in the manuscript only the 62 sample actually investigated, as the information on the 2 samples with haemolysis and not considered for the analyzes could be misleading.
As suggested by the Reviewer, we removed the 2 samples with haemolysis from the study.
In addition, information concerning the expression level of investigated miRNAs in serum from healthy donors could be very
We thank the Reviewer for this comment that led to the improvement of our study. In the short time that is allowed by MDPI for the revision, we collected and processed ten (10) normal blood plasma samples. Next all studied miRNAs were additionally quantified in these 10 normal samples. We added the respective information in the Materials and Methods, Results (including a revised Table S1), and Discussion.
Table S1: Distributions of the levels of circulating let-7b-5p, miR-143-3p, miR-17-5p, miR-214-3p and miR-335-5p in plasma samples of MM patients and normal controls.
- The information in the text and in the corresponding tables is quite redundant; in my opinion the paragraphs could be shortened.
Taking into consideration the Reviewer’s comment, we removed redundant information from the Results section.
- In figure 4b, the Kaplan Meier reported only 54 out of the 61 patients with clinical data, why?
As now mentioned in Table S1, miR-335-5p was quantified in 55 out of 62 plasma samples. Moreover, survival data of 1 out of these 55 patients are missing (as described in the section “3.4”: Of the total 62 MM patients, one was excluded from survival analysis due to missing follow-up data. From the rest 61 MM patients, …). Thus, the survival analysis based on circulating miR-335-5p levels was performed in 54 patients.
The authors wish to thank the Reviewers for their constructive comments that led to the improvement of the current manuscript.

Reviewer 2 Report
The authors propose a miRNA signature to facilitate MMBD diagnosis. The authors identified et-7b-5p and miR-335-44 5p as non-invasive prognostic biomarkers in MM.
The paper is very interesting and well done. However, I have some criticisms:
- The authors used plasma samples instead of serum samples. Some papers reported the use of serum samples. Why did the choose plasma? Did they check the differences between plasma and serum?
- Did the authors confirm the results obtained with peripheral samples in bone marrow samples? Please provide the comparison for some samples.
- Did the levels of circulating miRNAs correlate with the levels of bone markers such as alkaline phosphatase, CTX, NTX, and PINP or cytokines sRANKL, OPG, DKK-1, Sclerostin, ?
- Which is the predictive role of such miRNAs for fracture risk? Although challenging, the authors should include some controls (MGUS, SMM patients) and monitor the levels of miRNAs in the same patient before and after Osteolysis onset.
- How the levels of miRNAs change after therapy with bisphosphonate or other anti-resorptive therapy? Are their levels affected by treatment?
Author Response
The authors used plasma samples instead of serum samples. Some papers reported the use of serum samples. Why did the choose plasma? Did they check the differences between plasma and serum?
It is well documented in the existing literature that the selection of whole blood or particular blood fraction (plasma or serum) is a critical pre-analytical parameter, as it affects the miRNA profile. To the best of our knowledge, plasma is recommended over serum for circulating miRNA profiling, because the coagulation process modifies RNA content due to the activation of RNases released by activated platelets and also leads to the release of an excess amount of transfer RNA-derived RNA fragments. As a result, these reasons prompted us to investigate the levels of circulating miRNAs exclusively in blood plasma samples.
- Did the authors confirm the results obtained with peripheral samples in bone marrow samples? Please provide the comparison for some samples.
Promepted by the Reviewer’s suggestion, we performed additional experiments to investigate the expression levels of the five studied miRNAs in bone marrow samples. Thus, we included 30 bone marrow aspiration samples, 17 of which had previously been collected by patients with MMBD and 13 of which had previously been collected by patients without MMBD). We used these samples to isolate CD138+ plasma cells and performed miRNA quantification in the respective RNA extracts. Thus, we were able to provide comparison(s) for each miRNA. We added the respective information in the Materials and Methods, Results (including the new Figure S1), and Discussion.
Page 3, lines 108-111 (Materials and Methods): Peripheral blood plasma samples were obtained from 10 healthy donors as well as 62 multiple myeloma (MM) patients with or without MMBD at the time of diagnosis. Additionally, bone marrow aspirate samples (BMA) were obtained from 30 out of the 62 MM patients.
Page 3, lines 124-125 (Materials and Methods): Whole blood samples as well as BMA samples were collected in ethylenedia-minetetraacetic acid (EDTA) containing tubes from participants enrolled in the current study.
Page 3, lines 135-140 (Materials and Methods): Processing the BMA samples, we used Ficoll-Paque to isolate mononuclear cells. Next, CD138+ plasma cells were positively selected using magnetic beads coated with an anti-CD138 antibody (Miltenyi Biotec, Bergisch Gladbach, Germany). Total RNA extraction from CD138+ cell pellets was performed using TRI Reagent® (Molecular Research Center, Inc., Cincinnati, OH, USA). Finally, RNA pellets from bone marrow plasma cells were dissolved in THE RNA Storage Solution (Ambion™).
Page 4, lines 154-155 (Materials and Methods): Total RNA was also extracted from the CD138+ plasma cells positively selected from the BMA samples.
Page 4, lines 173-180 (Materials and Methods): Subsequently, we applied the comparative Cq method for the relative quantification of target miRNAs in the plasma samples from MM patients and normal controls, as well as in the CD138+ plasma cells from bone marrow of MM patients. For this purpose, we used reference transcripts to normalize qPCR for the amount of RNA used in RT reactions. Spiked-in cel-miR-39-3p was used for the normalization of circulating miRNA levels, whereas small nucleolar RNA C/D box 48 SNORD48 (RNU48) was exploited as an endogenous reference for the normalization of intracellular miRNA expression levels of CD138+ plasma cells.
Page 6, lines 238-241 (Results): Interestingly, a similar trend was observed for each of these miRNAs in CD138+ plasma cells that were positively selected from BMA samples of MMBD patients, compared to their intracellular expression levels in CD138+ plasma cells from BMA samples of MM patients without bone disease (Figure S1).
Page 13, lines 372-378 (Discussion): Α similar trend was also observed with regard to the intracellular levels of these five miRNAs, once their expression in CD138+ plasma cells of BMA samples was compared between MMBD patients and MM patients without bone disease. These results indicate that the levels of these circulating miRNAs in plasma could reflect their levels in ma-lignant cells of bone marrow. As a result, miRNA-based diagnostics may provide val-uable clinical information regarding osteolytic bone disease in MM.
Figure S1: Comparison of intracellular let-7b-5p, miR-143-3p, miR-17-5p, miR-214-3p, and miR-335-5p expression levels in CD138+ plasma cells from BMA samples of MM patients with and without MMBD.
- Did the levels of circulating miRNAs correlate with the levels of bone markers such as alkaline phosphatase, CTX, NTX, and PINP or cytokines sRANKL, OPG, DKK-1, Sclerostin?
We would like to thank the Reviewer for the comment. The only bone marker among the suggested ones for which we have collected data in our database is alkaline phosphatase (ALP). Therefore, as suggested by the Reviewer, we search for an association between ALP and circulating miRNA levels; no such correlation was observed, as mentioned in the Results of the revised manuscript:
Pages 7-8, lines 249-254 (Results): Although the aforementioned miRNAs can predict osteolytic bone disease in MM patients, our results failed to establish any significant association between their levels in plasma and the severity of MMBD as indicated by the number of osteolytic lesions and the presence of SREs. Moreover, no significant correlation was observed between the circulating levels of these miRNAs in plasma and serum alkaline phosphatase (ALP) levels.
- Which is the predictive role of such miRNAs for fracture risk? Although challenging, the authors should include some controls (MGUS, SMM patients) and monitor the levels of miRNAs in the same patient before and after Osteolysis onset.
Unfortunately, we do not have blood plasma samples collected from the MM patients once they were diagnosed with monoclonal gammopathy of undetermined significance (MGUS) or smoldering MM (SMM). Time intervals between MGUS or SMM and MM are usually long. Moreover, out of the 27 non-MMBD patients included in the study, only 2 cases showed osteolytic lesions during the follow-up period. Therefore, a much longer follow-up period is needed, in order to have such events.
For the same reason, we decided to include blood plasma samples of normal controls. Therefore, we observed that the levels of these five circulating miRNAs are quite similar with those in plasma of MM patients not presenting with osteolysis. We added the respective information in the Materials and Methods, Results (including a revised Table S1), and Discussion.
Table S1: Distributions of the levels of circulating let-7b-5p, miR-143-3p, miR-17-5p, miR-214-3p and miR-335-5p in plasma samples of MM patients and normal controls.
- How the levels of miRNAs change after therapy with bisphosphonate or other anti-resorptive therapy? Are their levels affected by treatment?
We agree with the Reviewer that it would be tempting to conclude whether bisphosphonate or other anti-resorptive therapy have any effect on the circulating levels of these miRNAs. However, as we do not have blood samples collected at particular time points, we cannot perform such analysis.
The authors wish to thank the Reviewers for their constructive comments that led to the improvement of the current manuscript.

Round 2
Reviewer 2 Report
Thank you for accomodating my requests